# The dynamic and structural properties of axonemal tubulins support the high length stability of cilia

Ron Orbach [1] & Jonathon Howard[1]

Cilia and flagella play essential roles in cell motility, sensing and development. These organelles have tightly controlled lengths, and the axoneme, which forms the core structure, has exceptionally high stability. This is despite being composed of microtubules that are often characterized as highly dynamic. To understand how ciliary tubulin contribute to stability, we develop a procedure to differentially extract tubulins from different components of axonemes purified from *Chlamydomonas reinhardtii*, and characterize their properties. We find that the microtubules support length stability by two distinct mechanisms: low dynamicity, and unusual stability of the protofilaments. The high stability of the protofilaments manifests itself in the formation of curved tip structures, up to a few microns long. These structures likely reflect intrinsic curvature of GTP or GDP·Pi tubulin and provide structural insights into the GTP-cap. Together, our study provides insights into growth, stability and the role of post-translational modifications of axonemal microtubules.

[1] Department of Molecular Biophysics and Biochemistry, Yale University, New Haven, CT, USA. Correspondence and requests for materials should be addressed to J.H. (email: jonathon.howard@yale.edu)

Eukaryotic cilia and flagella, terms often used interchangeably, are evolutionarily conserved hair-like organelles[1] that play key roles in cell motility, mechanical and chemical sensing, and development[2,3]. The core structure of the cilium, the axoneme, has a 9 + 2 microtubule (MT) architecture: two singlet MTs, known as the central pair (CP), and nine surrounding doublet MTs[4,5]. The singlet MTs are hollow cylindrical structures formed by 13 protofilaments (PFs) of the αβ-tubulin heterodimers. The doublet MTs consist of the A-tubule, which forms a complete MT with 13 PFs, and the B-tubule, which is attached to the A-tubule, and forms an incomplete MT structure with 10 PFs[6]. Interestingly, the MTs in the axoneme are very stable against depolymerization compared to cytoplasmic MTs. Behnke & Forer identified four classes of MTs based on their resistance to high temperature: cytoplasmic MTs < CP < B-tubule < A-tubule[7]. This result was further supported by another study using a surfactant to depolymerize doublet MTs[8].

Non-axonemal MTs are known as dynamic polymers whose ends switch stochastically between phases of slow growth and rapid shrinkage, in a process called dynamic instability. The switch from growth to shrinkage is called catastrophe and the reverse switch from shrinkage to growth is called rescue[9]. This dynamic nature of MTs is a key element that drives various cellular processes including cell architecture, division, and motility[10–12]. In cilia, however, dynamic instability is suppressed, and instead, assembly and disassembly are in a steady state, through mechanisms that are not understood, leading to remarkable length stability[13].

Here, we ask whether the tubulins from the various axoneme components have different dynamic and functional properties that contribute to the length stability of axonemes. One of the important questions we address is the role of post-translational modifications (PTMs). Tubulin is subjected to various PTMs, such as acetylation, polyglutamylation and detyrosination[14–16], and in cilia PTMs have been proposed to regulate length[17], stability[18–21], and motility[22–24]. However, while we already know much about their role in regulating MT interaction with other microtubule-associated proteins (MAPs)[15,16], the role of PTMs on MT dynamics has remained elusive. This is partially due to the fact that most studied organisms have multiple tubulin isotypes, which may provide another source of functional diversity, and together with PTMs compose the tubulin code.

We decided to use the unicellular biflagellate green alga *Chlamydomonas reinhardtii* to study ciliary MTs in part because any molecular heterogeneity of their MTs is due only to PTMs. *C. reinhardtii* has two genes that encode α-tubulin and two genes that encode β-tubulin[25]. Each pair of genes specifies an identical protein product and therefore only a single type of tubulin heterodimer is synthesized[26,27]. Yet, a major challenge is the difficulty in purifying functional tubulins from the axoneme. Previous studies used harsh conditions, such as ionic detergent[8], high temperature or proteases[7] to extract tubulins from the axoneme structure, and therefore led to denaturation of the tubulins. Other studies used sonication to purify axonemal tubulin from sea urchin, however, only small amounts of tubulin could be extracted this way and not in a manner that allows differential extraction of tubulin from different axoneme components[28,29].

In this study, we present a procedure to purify functional tubulins from the different components of the axoneme, which allows us also to determine PTMs distribution within the axonemal structure. We characterize the dynamic properties of axonemal MTs and find that the MTs support length stability of the axoneme by low dynamicity, and high stability of the PFs. Interestingly, we find that the stable PFs form long curved tip structures at the very end of growing MTs.

## Results

**Purification of axonemal tubulin.** To overcome the challenge of extracting functional tubulins, we decided to use the Hofmeister series[30], which classifies the effect of different ions on the solubility and stability of proteins[31]. We searched for a salt that destabilizes the axoneme structure but does not cause tubulin denaturation and the loss of its ability to repolymerize. Purified axonemes were incubated with different salts from the Hofmeister series and subsequently pelleted (Fig. 1a). Depolymerized tubulin and other soluble proteins remained in the supernatant, while intact axonemes and other insoluble proteins sedimented into the pellet. Based on this assay, we decided to test $NaNO_3$ and KI, which both solubilize significant amounts of tubulin. We found that the more chaotropic salt, KI, leads to tubulin denaturation and no tubulin polymerization was observed after treatment with this salt. In contrast, $NaNO_3$ has no significant effect on the dynamic properties (Supplementary Fig. 1), and therefore was selected for our purifications.

We exploited the limited extraction of tubulin by $NaNO_3$ to sequentially extract tubulins of different stabilities. The axonemes were subjected to five consecutive cycles of salt extraction (Fig. 1b). Interestingly, even after five cycles of extraction, with a decreasing amount of tubulin extracted in each cycle, there was a significant amount of tubulin in the pellet. Intrigued by this observation, we examined the 9 + 2 architecture of the axoneme during the extraction process by transmission electron microscopy (TEM). After one cycle of extraction, the CP and a small part of the B-tubule were absent. After two cycles, another part of the B-tubule was missing, and after five cycles of salt extraction primarily A-tubules were observed in the pellet (Fig. 1c). Thus, different components of the axoneme structure exhibit various stabilities to salt extraction. The extracted tubulin was further purified by two chromatography steps - anion exchange and size exclusion - to separate tubulin from other MAPs (Fig. 1d). Finally, the functionality of the purified tubulin (SE fraction in Fig. 1d) to polymerize into MTs was confirmed by TEM (Fig. 1e). Tubulin purified from the combined cyc1-cyc5 fractions is referred from this point as CP/B tubulin.

**Axonemal MTs have distinct dynamic properties.** We characterized the dynamic properties of the various axonemal tubulins using interference reflection microscopy (IRM) to perform label-free dynamic MT assays[32]. Purified tubulin was assembled onto GMP-CPP stabilized MT seeds polymerized from bovine brain tubulin and attached to the coverslip (Fig. 2a, b). Table 1 summarizes the dynamical properties of the CP/B tubulin.

We first investigated how the growth rates depend on tubulin concentrations. The minimum concentration for elongation from the seeds was ~6 μM tubulin, similar to that in bovine brain tubulin. In comparison to bovine brain MTs, however, the growth rate at the plus end of axonemal MTs was significantly slower ($0.15\ \mu m\ min^{-1}$ vs. $0.30\ \mu m\ min^{-1}$) (Fig. 2c). We then performed linear regression on the growth rate vs. tubulin concentration (Fig. 2d) and found that the extrapolated tubulin concentration for growth at the plus end (x-intercept) was $1.35 \pm 0.77\ \mu M$, not significantly different from zero, and similar to bovine brain tubulin[33]. The slope of the axonemal tubulin curve corresponds to a second-order association rate constant of $0.54\ \mu M^{-1}\ s^{-1}$. This rate is significantly lower than previous measurements for brain tubulin[34] (after correcting for temperature). Thus, axonemal MTs have slower growth rates than bovine brain MTs.

To determine the differences between the two ends of the axonemal MTs we performed the same analysis for the minus end. As for the plus end, we found that the rate of polymerization was proportional to the tubulin concentration (Fig. 2e). Linear

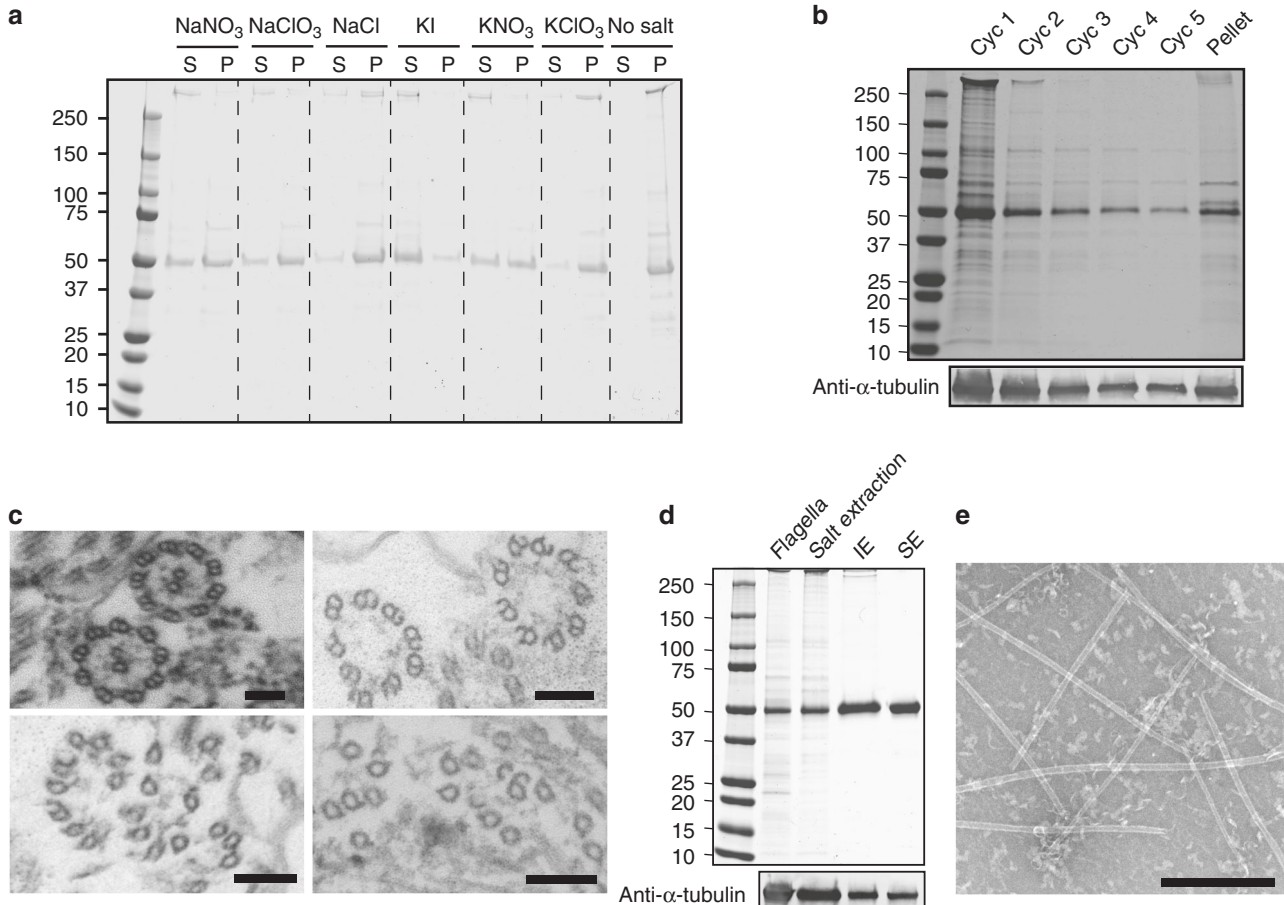

**Fig. 1** Purification of axonemal tubulin and MTs reconstitution. **a** SDS-PAGE of supernatant (S) and pellet (P) axoneme samples after salt extraction (500 mM). Supernatant contains soluble axonemal tubulin (50 kDa), while the pellet contains tubulin from insoluble axonemes. **b** SDS-PAGE of supernatant samples from five cycles of 500 mM $NaNO_3$ salt extraction, and the pellet at the end. **c** Transmission electron microscopy micrographs of intact axoneme (top left) and axonemes after 1 cycle (top right) 2 cycles (bottom left) and 5 cycles (bottom right) of salt extraction with 500 mM $NaNO_3$. Scale bar: 100 nm. **d** SDS-PAGE of samples collected throughout the axonemal tubulin purification. From left to right: Flagella lysate, supernatant of the axonemes after treating them with 500 mM $NaNO_3$ salt extraction, Hi-Trap Q (IE) elution fractions, size exclusion (SE) elution. **e** TEM micrograph of taxol-stabilized reconstituted axonemal MTs. Scale bar: 500 nm. Source data are provided as a Source Data file

regression showed that the x-intercept at the minus end is ~2-fold higher (2.58 ± 0.67 μM) compared to the plus end (Fig. 2f), and the slope of the curve corresponds to a second-order association rate constant of 0.43 $μM^{-1} s^{-1}$, only ~20% less than for the plus end. Thus, these results differ from brain MTs that show a 2-fold difference in the growth rates between the plus and minus ends[35].

We then interrogated the catastrophe rate, which may play an important role in the axoneme length stability. The catastrophe rate at the plus end was measured for different tubulin concentrations and compared to bovine brain MTs. The rate of catastrophe decreased as the concentration of tubulin increased, and compared to bovine brain MTs, axonemal MTs showed a 2-fold lower catastrophe rate (0.06 ± 0.01 $min^{-1}$ vs. 0.11 ± 0.01 $min^{-1}$, mean ± SEM) (Fig. 2g). Catastrophe events at the minus end, and rescue events at both ends were rarely detected in our assay, and for this reason, we omitted these parameters from the subsequent analysis.

To test whether the depolymerization (shrinkage) rate of axonemal MTs is different than bovine brain MTs, we followed it by IRM fast imaging (Fig. 2h). The mean shrinkage rate of axonemal MTs was 1.5 times faster than that of bovine brain MTs at the same tubulin concentration (9.93 ± 1.87 μm/min vs. 6.64 ± 0.72 μm/min, mean ± SEM) but not statistically significant.

Finally, we determined the dynamicity of axonemal and bovine brain MTs from the dynamic assay (Fig. 2i). Dynamicity was

calculated as previously reported by the sum of total growth and shrinkage lengths divided by the total time[36]. There was a 2-fold decrease in dynamicity of axonemal MTs compared to bovine brain MTs (0.14 ± 0.02 μm/sec vs. 0.28 ± 0.03 μm/sec, mean ± SEM). Thus, axonemal MTs are less dynamic than bovine brain MTs.

**PTMs have no effect on dynamic properties of axonemal MTs.** We further investigated any possible dynamic differences between CP and doublet tubulins by purifying tubulin from the separate CP and the B-tubule fractions. Because the CP MTs are known to be more labile than the doublet MTs[7], we developed a protocol for differential extraction, in which one extraction at 330 mM $NaNO_3$ was followed by four extractions at 500 mM $NaNO_3$. Consequently, the first cycle included CP tubulin (TEM images confirmed lack of CP, Supplementary Fig. 2), while the other four cycles included mainly tubulin from the B-tubule, and the pellet comprised A-tubule MTs. Thus, we were able to solubilize tubulin and to get highly enriched CP or B-tubule tubulin samples.

Previously it was reported that axonemal MTs from the CP, B-tubule and A-tubule have different PTMs. Hence, we analyzed the PTMs of the extracted tubulins in the different cycles by immunoblot (Fig. 3a–b). Anti-α-tubulin antibody verified that similar amounts of tubulin were found in the pellet and the first

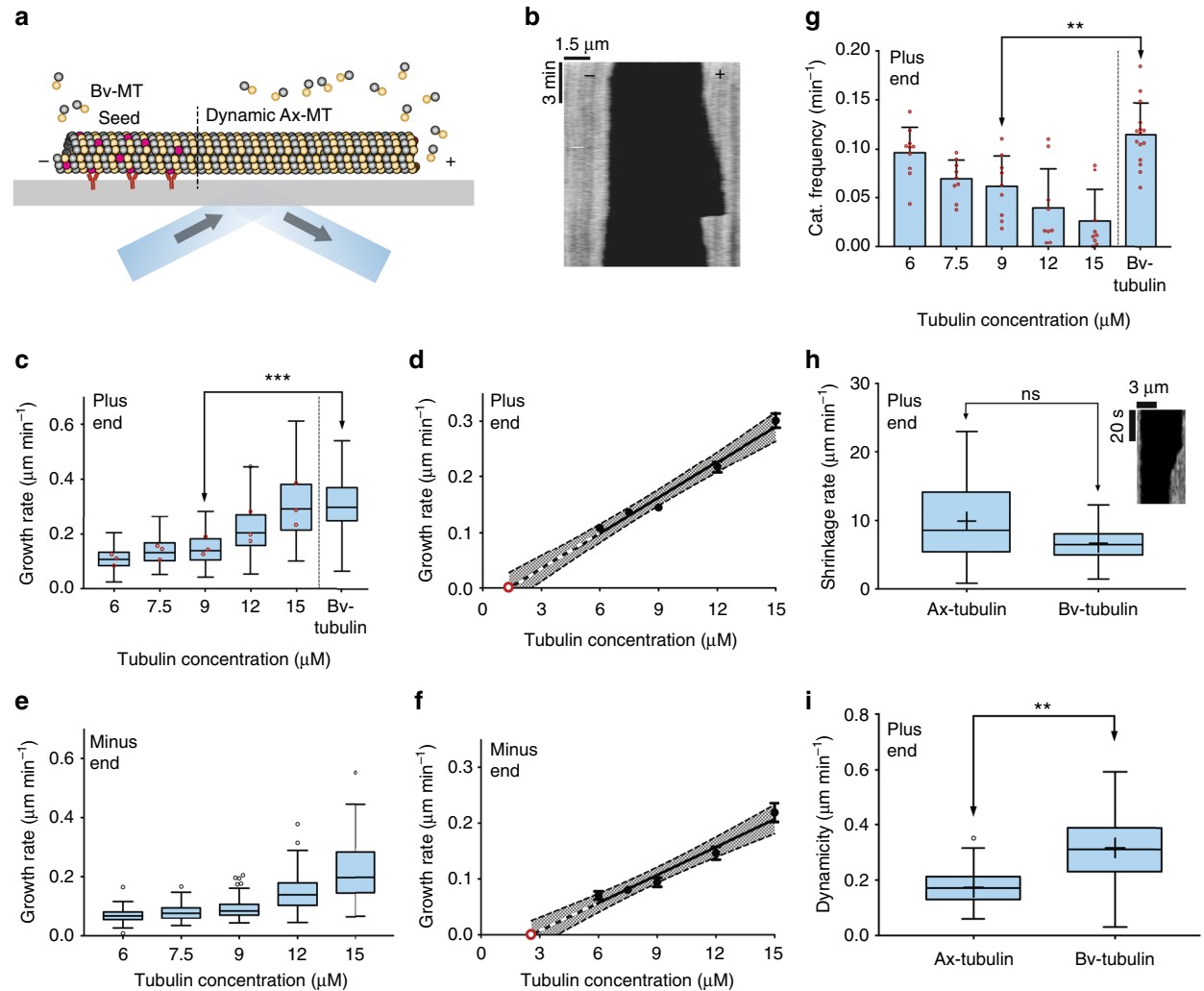

**Fig. 2** Polymerization dynamics of axonemal microtubules. **a** Schematic of the experimental design. Axonemal MTs grow from GMPCPP-stabilized TAMRA-labeled bovine brain MT seeds, which are immobilized to the coverslip with anti-TAMRA antibodies. **b** Representative kymograph of axonemal MT growth imaged using interference reflection microscopy. **c** Tukey plot showing growth rates of axonemal MTs in different tubulin concentrations. Each red circle marks the mean growth rate of a single biological replicate ($n = 275, 280, 268, 252, 265, 542$ total number of MTs used at the different concentrations; ***$p = 0.0003$, determined by a two-tailed t test, 9 independent experiments for each tubulin concentration from three independent axonemal tubulin purifications). **d** The mean growth rates at different tubulin concentrations from panel (c) were fit by linear regression. No growth is observed below 6 μM and the x-intercept (red open circle) is 1.35 μM. Grey area denotes 95% confidence interval. Error bars represent SEM. **e** Tukey plot showing growth rates at the minus end of axonemal MTs in different tubulin concentrations ($n = 48, 60, 73, 96, 126$). **f** The mean growth rates at different tubulin concentrations from panel (e) were fit by linear regression. The extrapolated concentration is 2.6 μM. Grey area denotes 95% confidence interval. **g** Bar plot showing catastrophe frequency of axonemal MTs at different tubulin concentration and bovine brain tubulin at 9 μM (Error bars represent SD for $n = 10, 9, 9, 9, 9, 15$; **$p = 0.0027$, determined by a two-tailed t test). **h** Tukey plot showing the shrinkage rates of axonemal MTs and bovine brain MTs at 9 μm tubulin concentration ($n = 9$ independent experiments with a total of 111, 211 MTs). Inset: kymograph showing a typical shrinkage event. **i** Tukey plot showing the dynamicity of axonemal MTs and bovine brain MTs at 9 μM tubulin concentration (a total of 75 MTs were analyzed in each of the two tubulins; **$p = 0.001$, determined by a two-tailed t test, 9 independent experiments for each tubulin concentration). For all Tukey plots, middle line represents the median, plus sign indicates the mean. Source data are provided as a Source Data file

**Table 1 Rate constants for axonemal tubulin**

|  | Plus end | Minus end |
|---|---|---|
| Slope (μm μM⁻¹·min⁻¹) | $0.021 \pm 0.001$ | $0.016 \pm 0.001$ |
| $k_{on}$ (μM⁻¹ s⁻¹) | $0.54 \pm 0.03$ | $0.43 \pm 0.03$ |
| $y$-intercept (μm min⁻¹) | $-0.029 \pm 0.011$ | $-0.043 \pm 0.012$ |
| $k_{off}$ (s⁻¹) | $0.785 \pm 0.297$ | $1.164 \pm 0.325$ |
| $x$-intercept (μM) | $1.35 \pm 0.77$ | $2.58 \pm 0.67$ |

Values represent mean ± SEM

two cycles of extraction, whereas in each of the following cycles the amount of extracted tubulin decreased. The extracted tubulin in the first cycle and the pellet had reduced polyglutamylation, while the tubulin from the other four cycles showed higher amounts of polyglutamylation. These results are consistent with a previous study that showed that MTs in the CP and the A-tubule have a reduced amount of polyglutamylation compared to the B- tubule[37]. Tubulins from the CP, B-tubule and A-tubule were all acetylated, and contained a mixture of tyrosinated, detyrosinated and Δ2-tubulin. We then examined the glycylation of the axoneme. TAP952 and AXO49 antibodies, which are frequently used to detect glycylation in motile cilia, did not label

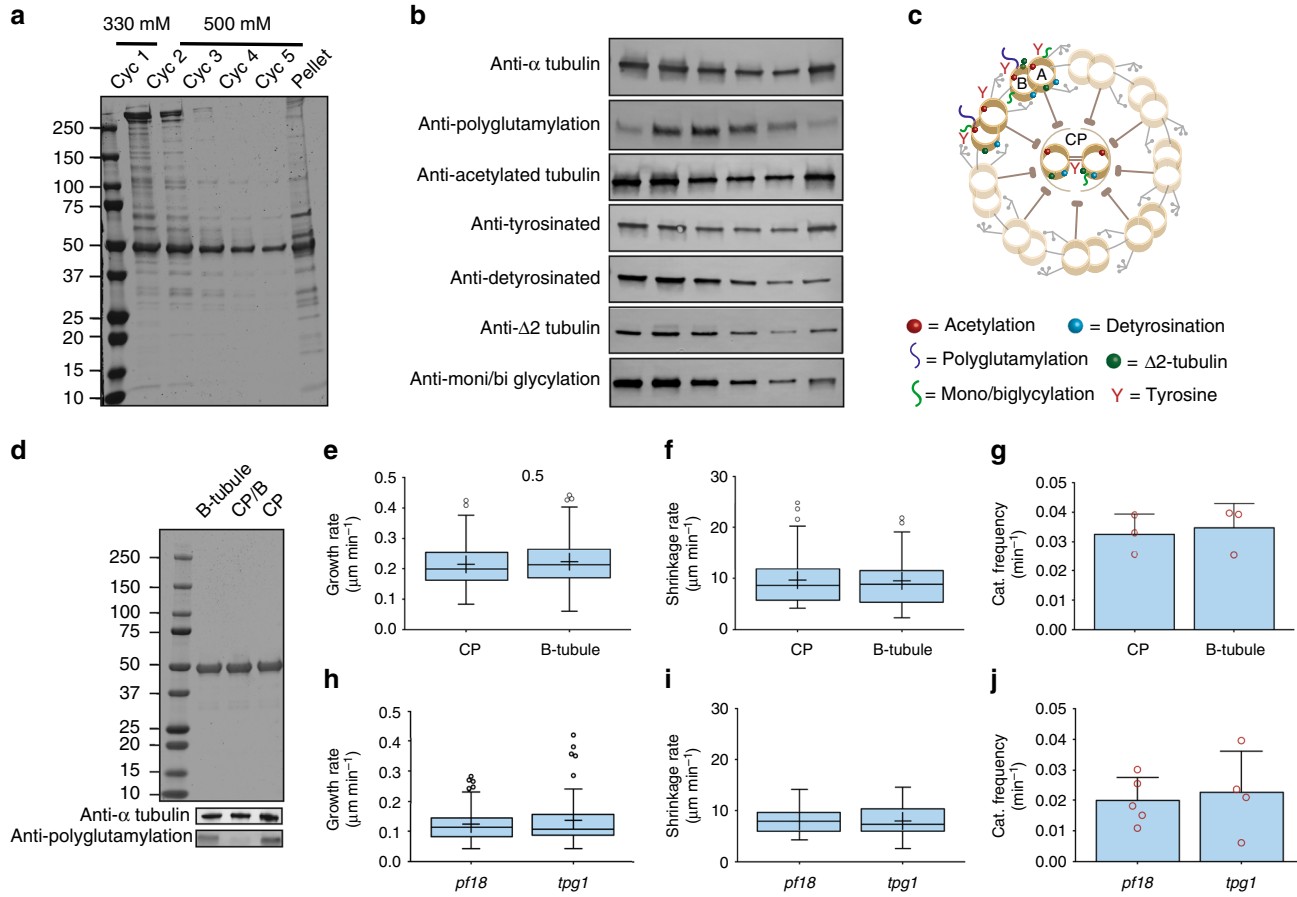

**Fig. 3** Purification, post-translational modifications analysis and dynamics. **a** SDS-PAGE of samples extracted from the axoneme with lower NaNO₃ concentration in the first cycle only. **b** Immunoblots of samples from the salt-extraction cycles. **c** Schematic of the different PTMs on the axoneme structure. **d** SDS-PAGE and immunoblot of axonemal tubulin purified from the CP, B-tubule and combined fractions (CP/B). **e** Tukey plots showing growth rate (127, 146 MTs) and **f** shrinkage rate of tubulin from the CP and B-tubule (52, 50 MTs) at 9 μM tubulin concentration. Middle line represents the median, plus sign indicates the mean. **g** Average catastrophe frequency of axonemal MTs polymerized from CP and B-tubule at 9 μM tubulin concentration. Error bars are SD for $n = 3$ independent experiments. **h** Tukey plots showing growth rate (82, 201 MTs) and **i** shrinkage rate of tubulin from the *pf18* and *tpg1* at 9 μM tubulin concentration (24, 25 MTs). Middle line represents the median, plus sign indicates the mean. **j** Average catastrophe frequency of axonemal MTs polymerized from *pf18* and *tpg1*, derived from three independent experiments. Error bars are SD for $n = 5$, 4 independent experiments. Source data are provided as a Source Data file

any of the tubulins. However, Gly-Pep1, which labels glycylated tubulin in primary cilia[17], labeled the tubulin in the CP and B-tubule, and to a lesser extent tubulin from the A-tubule. Thus, our purification assay provides several insights into the distribution of the PTMs that decorate the axonemal MTs (Fig. 3c and Supplementary Fig. 3 and Fig. 4).

To determine whether these PTMs have an effect on dynamics, we further purified the tubulins from the CP, and the B-tubule (Fig. 3d). No significant differences were observed for growth rate (Fig. 3e), shrinkage rate (Fig. 3f), or catastrophe frequency (Fig. 3g). These results were supported by the characterization of the dynamic properties of two other *C. reinhardtii* mutants: (1) *pf18* that completely lacks the CP so that most of the purified axonemal tubulin consists of polyglutamylated tubulin; and (2) *tpg1* that lacks tubulin tyrosine ligase-like 9 (TTLL9) and has a reduced amount of polyglutamylation[22]. In both strains no significant differences were observed in their dynamic properties (Fig. 3h–j). Thus, our experiments with a single isotype tubulin indicate that there are no detectable differences in the dynamic properties between CP and B-tubule tubulins.

**Axonemal MTs form curved tip structures.** During the analysis of the reconstituted axonemal MTs we observed a unique

phenomenon for all tubulins (CP, B-tubule and CP/B). At tubulin concentrations above 9 μM, 5–10% of the growing MTs formed curved tip structures (Supplementary Movie 1), not observed for brain MTs (Supplementary Movie 2). The tips were dim compared to the MT shaft, indicating a lower number of PFs (Fig. 4a), and showed a range of lengths and curvatures (Fig. 4b). The mean tip length of ~1.5 μm was independent of tubulin concentration, while the tip curvature (equal to 1/radius of curvature) increased from $2.0 \pm 0.8$ μm⁻¹ at 9 μM to $2.7 \pm 1.1$ μm⁻¹ at 15 μM (mean ± SEM) (Table 2 and Supplementary Fig. 5a–b). The distal tip curvature showed a small but significant decrease with tip length (Supplementary Fig. 5c). We further examined the tips of axonemal MTs by TEM and found open sheets at the MT ends (Fig. 4c). We therefore hypothesized that the tip structures are unclosed sheets comprising several PFs.

This hypothesis is consistent with several unusual growth features seen for MTs with tip structures. First, the tips darkened as they straightened (Supplementary Movie 1), consistent with a transition from a curved sheet to a straight MT. Second, we observed fast growth phases of the MTs (Fig. 4d, Supplementary Movie 3), including after catastrophe events (Fig. 4e, Supplementary Movie 4). These observations are consistent with incomplete tip structures providing a template for rapid MT

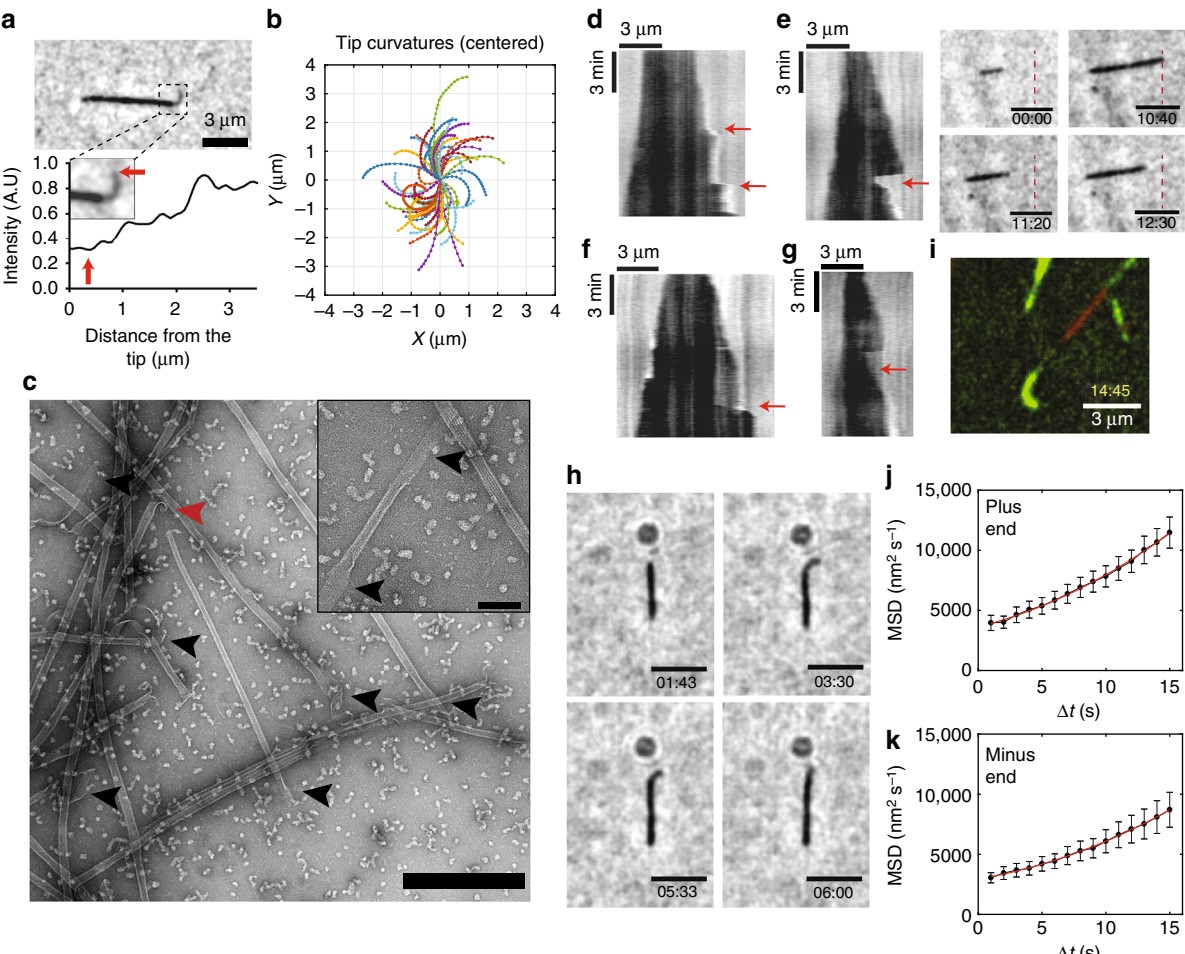

**Fig. 4** Growth of curved protofilaments polymerized from axonemal tubulins. **a** An image of curved PFs (dim region) growing from a MT (top). Intensity profile of the MT tip shows stepwise increases (bottom). Red arrow indicates the tip. **b** Traces of individual curved PFs. **c** TEM image of growing MTs with open sheets at the ends (indicated in black arrows), and shrinking MT with ram's horns (red arrow). Scale bar: 500 nm. Inset: high magnification of MT with open sheets at its ends. Scale bar: 100 nm. **d** Kymograph showing fast growth phases. **e** Kymograph and images showing fast growth phase after catastrophe event (dashed red line as reference). **f** Kymograph showing the growth of a few PFs (dim region) and the growth of additional PFs alongside them. **g** Kymograph showing stable PFs after catastrophe event. Red arrows indicate each event. **h** Images showing the formation of curved tip structure following a catastrophe event. **i** TIRF microscopy image showing a curved tip structure (see also Supplementary Movies 9 and 10). **j**, **k** Mean-squared displacement (MSD) against time. A quadratic fit yielded a diffusion coefficient and velocity at the plus and minus ends

**Table 2 Structural properties of axonemal curved tips**

|  | 9 μM | 12 μM | 15 μM |
|---|---|---|---|
| Number of MTs (n) | 43 | 49 | 42 |
| Mean length (μm) | 1.53 ± 0.07 | 1.36 ± 0 .07 | 1.67 ± 0.08 |
| Min length (μm) | 0.61 | 0.65 | 0.65 |
| Max length (μm) | 2.87 | 2.69 | 3.32 |
| Length SD (μm) | 0.48 | 0.51 | 0.55 |
| Mean curvature (μm$^{-1}$) | 1.95 ± 0.12 | 2.25 ± 0.13 | 2.71 ± 0.17 |
| Curvature SD (μm$^{-1}$) | 0.77 | 0.92 | 1.13 |

Values represent mean ± SEM

growth. Third, we observed rapid growth along dim tip structures (Fig. 4f, Supplementary Movie 5). Fourth, occasionally we identified dim regions of the MTs after catastrophe events, suggesting that only some PFs underwent catastrophe (Fig. 4g, Supplementary Movie 6). This observation was confirmed by fast imaging of MTs that form a dim curved tip after a catastrophe event (Fig. 4h, Supplementary Movies 7 and 8), and by similar observations using total internal reflection fluorescence (TIRF)

microscopy (Fig. 4i, Supplementary Movies 9 and 10). Taken together, these observations suggest that PF sheets provide stable structures that allow fast templated growth.

Though the curved tips were only observed in 5–10% of MTs imaged by IRM, we wondered whether they might be present in a larger fraction of MTs, but not seen due to their shorter lengths. One characteristic of growth in the presence of curved tips is the variability of growth velocities. To determine whether variable growth is a property of all the axonemal MTs, we tracked the position of MT ends without observed PFs, and performed a mean-squared displacement (MSD) analysis in which the average of the square of the displacement in a time interval is plotted against the time interval (Fig. 4j–k). By fitting our results to a quadratic equation, we calculated the average growth rate (*V*) and the diffusion coefficient (*D*), which is a measure of the variability of the growth rate. The diffusion coefficient at the plus end is much larger than expected if the only source of growth variability is the random arrival of tubulin dimers that bind irreversibly at the end (Table 3). Thus, all MTs display growth variability, even those that do not display tip structures. Previously, such high diffusion coefficients have been interpreted as arising from reversible binding of tubulin dimers at the growing tip, so that a

**Table 3 Estimated rate constants from MSD analysis**

|  | Plus end | Minus end |
|---|---|---|
| Number of MTs ($n$) | 45 | 43 |
| Diffusion coefficient ($D$, nm$^2$ s$^{-1}$) | 130.26 ± 15.30 | 87.50 ± 10.39 |
| Growth velocity ($V$, nm s$^{-1}$) | 4.14 ± 0.22 | 3.72 ± 0.17 |
| Expected Diffusion coefficient$^a$ ($D^*$, nm$^2$ s$^{-1}$) | 2.55 ± 0.13 | 2.28 ± 0.10 |
| $k_{on}$[Tub] (s$^{-1}$) | 347.76 ± 40.41 | 234.36 ± 27.45 |
| $k_{on}$ ($\mu$M$^{-1}$ s$^{-1}$) | 38.64 ± 4.49 | 26.04 ± 3.05 |
| $k_{off}$ (s$^{-1}$) | 341.03 ± 40.46 | 228.33 ± 27.47 |
| $\sigma^2$ (nm$^2$) | 60.33 ± 0.88 | 54.32 ± 0.66 |

Values represent mean ± SEM
$^a$Diffusion coefficient assuming random and irreversible tubulin dimer binding, $D^* = V \times 8$ nm/13 PFs

majority of the tubulins that bind also unbind[34]. However, as we argue in the Discussion, the PF templated growth may provide as an alternative interpretation.

## Discussion

To understand the role of MTs in length stability of axonemes, we studied axonemal tubulin purified from *C. reinhardtii*. To extract both CP and doublet tubulins, we used salts from the Hofmeister series to destabilize the axonemes, and by adjusting the salt concentration of NaNO₃ we were able to purify single isotype functional tubulins from the CP and the B-tubule separately. We found that *C. reinhardtii* axonemal MTs show a 2-fold decrease in dynamicity compared to bovine brain MTs, which is reflected by a slower growth rate and lower catastrophe frequency. Thus, the high stability of the *C. reinhardtii* axoneme could be due, in part, to the reduced dynamical properties of its tubulin.

Axonemal MTs still undergoe dynamic instability, however, and we asked whether PTMs influence dynamics. Previous studies, which were based on mammalian brain tubulin, involved several tubulin isotypes and consequently the effect of PTMs on the dynamic properties has remained elusive. However, *C. reinhardtii* has a single tubulin isotype and therefore we can isolate this effect. We characterized the PTMs on the different axoneme components (CP, B-tubule and A-tubule). In agreement with a previous report[37], we found that polyglutamylation, which was suggested to regulate ciliary motility[22,24,38,39], was mainly in fractions enriched in B-tubule tubulin, and lower in fractions enriched in CP and A-tubule. Interestingly, our dynamic assays with purified tubulins from the CP and B-tubule, as well as two different *C. reinhardtii* mutants, showed that they have similar dynamic properties. Thus, polyglutamylation appears to have no effect on the dynamic properties of MTs.

Detyrosination, Δ2-tubulin and mono/biglycylation signals were all weaker in the A-tubule fraction than in the CP and B-tubule fractions. This suggests that both CP and B-tubule have these modifications, as the A-tubule fraction serves as a control for background signal. On the other hand, tyrosination and acetylation are similar for all fractions. However, the decrease in the detyrosination and Δ2-tubulin of the A-tubule imply that tyrosination should increase, since tubulin can have only one of these three states. One explanation for these results might be related to the high efficacy of the tyrosinated-tubulin antibody, which hinders the detection of small differences[40]. In the case of acetylation, we believe that the signal is indicating that tubulin from CP, B-tubule and A-tubule are all indeed acetylated, and to a similar extent. This is because control immunoblots with yeast tubulin (Supplementary Fig. 4), which is not acetylated, and *Chlamydomonas* and bovine brain tubulin treated enzymatically to remove acetylation[24], show much weaker staining with this antibody.

Thus, we conclude that detyrosination, Δ2-tubulin, mono/biglycylation and acetylation are all present on CP and B-tubule.

During the characterization of the axonemal MTs, we discovered that at high tubulin concentrations (>9 μM) some of the growing axonemal MTs have relatively dim, curved tips at their ends. We do not think that the tip structures are caused by contamination of MAPs because our purification scheme includes a stringent salt extraction followed by two chromatography steps, anion exchange and size exclusion. This was supported by mass spectrometry analysis, which failed to identify any known MAPs (Supplementary Tables 1 and 2), and SDS-PAGE with high tubulin load (15 μg) that showed only low molecular weight proteins, which are associated with tubulin degradation (Supplementary Fig. 6). Additionally, no significant change or trend in the depolymerization rate were observed at different tubulin concentrations (Supplementary Fig. 7); this argues against the presence of a MAP that modulates the shrinking rate. Finally, similar tip structures were observed in samples of axonemal tubulin that were cycled after purification (Supplementary Movies 11 and 12). The lower intensity and high curvature distinguish these tip structures from the cylindrical structure of complete MTs, and suggest that they are formed by unclosed PF sheets, which we also observed in TEM, and others have observed in cryo-EM[41–43]. Our failure to observe them for bovine brain MTs suggests that axonemal PFs may be more stable than those of brain tubulin.

Tip structures were detected primarily at high concentrations. However, some tip structures were also seen at low tubulin concentrations. The reason that fewer tip structures were seen at lower concentrations could be because the tip structures are shorter due to the lower growth rates. Thus, it is possible that the tip structures are more prevalent than we report. If this is the case, then tip structures could contribute to the high growth variability of the axonemal MTs as revealed by our MSD analysis (Table 3).

High PF stability may account for several unusual features of growth. For example, fast growth phases are observed, despite the low average dynamicity of the MTs. Moreover, the PFs show unique stability to catastrophe events, which is reflected by PFs that do not depolymerize after these events, and provide a template for newly polymerizing PFs. This provides a mechanism for fast recovery of the MTs from catastrophe events. While we observe these phenomena only in axonemal MTs, this type of behavior might happen to a lesser extent in other tubulins with much shorter tip structures as observed in some EM studies, and could explain the rapid fluctuations in their growth rates[34,44,45].

An important question is whether the differences in dynamics and PFs stability between *C. reinhardtii* axonemal MTs and bovine brain MTs are due to differences between ciliate and non-ciliate tubulins or to differences between axonemal and non-axonemal tubulins. One possible answer to this question might be related to differences in the lateral and longitudinal interactions. Previous studies showed that Lys-58 stabilizes the lateral contacts between adjacent subunits in bovine and porcine brain MTs[46,47]. However, in *C. reinhardtii* Lys-58 is substituted for Arg in β-tubulin. This can provide a longer side chain, and tolerance to structural changes that accompany the MT before catastrophe events[47] (Supplementary Fig. 8). Intriguingly, we found that Arg-58 is common among ciliates, whereas in other species that are commonly used to study MTs, Lys-58 is highly conserved (Supplementary Fig. 9). Because Lys-58 is acetylated in human tubulin[48], it is possible the Lys-58 substitution to Arg alters a potential regulatory site. Another substitution is Tyr-222 that is located at the E-site and interacts with the nucleobase and with Asp-179, which takes an active role in the formation of longitudinal interactions[49]. In *C. reinhardtii* Tyr-222 is substituted for

a Phe residue, which is missing the hydroxyl group and therefore cannot interact with Asp-179 (Supplementary Fig. 10), and might have an effect on any potential phosphorylation. This frees Asp-179 to promote longitudinal contacts. Another role of Tyr-222 is in the positioning of the GTP in the E-site. However, the interaction of Phe with the nucleobase might lead to different positioning and consequently affecting GTP hydrolysis. As in the other case, Tyr-222 is highly conserved between other species that are used to study MTs (Supplementary Fig. 11), while Phe-222 is found only in plants[50]. Thus, these residues may provide a structural basis for the functional differences between *C. reinhardtii* axonemal tubulin and bovine brain tubulin.

There are also sequence similarities between axonemal tubulins of non-ciliates and ciliates. Previously, it was demonstrated that tubulin isotypes with the 'EGEFXXX' motif in the C-terminal tail of β-tubulin, where X is an acidic amino acid, are enriched in cilia of *Drosophila*[51] and cows[52,53]. It was further suggested that these isotypes are required for ciliary function and assembly. We find that the 'EGEFXXX' motif is also found in ciliates (though in *C. reinhardtii* and *Volvox carteri* the second X is replaced with glycine) (Supplementary Fig. 12), supporting the notion that this motif may have a function in motile cilia from both ciliates and non-ciliates. Furthermore, it was previously demonstrated that the tubulin C-terminal tail is playing a role in regulating MT dynamics[54,55]. This might imply that the 'EGEFX(X/G)X' motif also contributes to the low dynamicity of axonemal MTs. The function of this motif may be a direct consequence of its sequence, or may be due to this motif having different PTMs compared to other C-terminal tails.

*C. reinhardtii* has a single tubulin isotype, which might suggest that cytosolic and axonemal MTs have similar dynamicity. However, previous study showed that cytosolic MTs are highly dynamic[56], therefore indicating that there are other mechanisms, such as MAPs or microtubule-inner proteins[57] or other PTMs, that differentially regulate the stability of the MTs in the axoneme and the cytosol.

A long-standing question concerns the mechanism of MT assembly by GTP-tubulin. Early observations of highly curved PFs at the MT tips during depolymerization and relatively straighter PFs during polymerization, paved the way to a canonical model[58]. According to this model, GTP-tubulin forms a straighter conformation compared to GDP-tubulin, which is more compatible with the MT lattice, and thus promotes the polymerization of MTs. The delayed hydrolysis of GTP to GDP after polymerization leads to a strained GDP-tubulin configuration that is forced to adopt an energetically unfavorable straight conformation within the MT lattice. This strain energy is released by catastrophe events that allow the GDP-tubulin to assume its preferred curved conformation[59]. Alternative models posit that tubulin has a curved conformation, independent of its nucleotide state[60]. This is based on the crystal structures of tubulin[49,61–63], and cryo-EM analyses that showed slightly curved PFs at the end of growing MTs[41–43,64,65]. These studies were further supported by biochemical experiments[61–63,66,67]. Thus, there is uncertainty about the extent to which GTP-tubulin adopts a curved PF structure.

Our dynamic assays provide a different insight into MT tip growth (Fig. 5), since all previous studies were based on static images. The assays indicate that growing MTs generate curved tip structures; if the tubulin in these structures is in the GTP state, as had been argued by Odde et al.[34], then our observations support a curved GTP-tubulin configuration. The radius of curvature at the very ends of the tips is ~400 nm, an order of magnitude larger than the tightly curved ram's horns observed in shrinking MTs that are thought to represent the GDP state[65]. These values are also much larger than the values previously reported by cryo-EM studies[41–43],

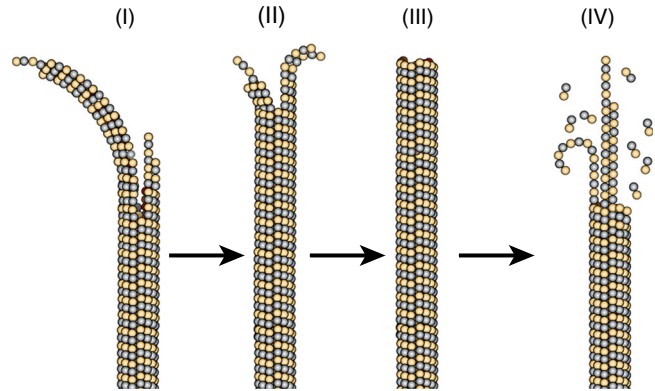

**Fig. 5** Model for growth and shrinkage for axonemal tubulin. (I) MTs polymerize by incorporation of GTP-tubulin at their tip. During this process PFs form curved tip structures with a radius of curvature of ~400 nm. (II) These curved PFs provide a template for polymerization of new PFs that catch up with the previous PFs, and thus leading to conformational change that decreases the curvature of the tip. (III) The straight and full MT is formed by the polymerization of 13 PFs. (IV) Upon catastrophe, a few PFs remain behind and provide a template for new PFs, while others can form ram's horns structure that has a much smaller radius of curvature

suggesting that the curved tip structure in our assays include higher number PFs. Future studies will include cryo-EM characterization that will provide a better resolution to axonemal MT tips.

In conclusion, *C. reinhardtii* provides a unique platform to study MTs structural and biophysical properties. Our results show that the dynamic properties of axonemal tubulin together with its structural properties provide two distinct mechanisms that can support the high length stability of the axoneme structure. It should be noted, however, that further studies will be required to determine explicitly whether these observations are also true to axonemal tubulin from other sources. The single tubulin isotype in *C. reinhardtii* allowed us also to study the effect of PTMs without the complexity of isotypic heterogeneity. Thus, we can conclude that polyglutamylation has no effect on the dynamic properties of *Chlamydomonas* axonemal tubulins. Finally, axonemal tubulin provides an insight into the growth of MTs and specifically the conformation of GTP-tubulin.

## Methods

***Chlamydomonas* strains and culture conditions**. *C. reinhardtii* wild-type strain 137c (CC-125), *pf18* (CC-1036) and *tpg1*(CC5245) were obtained from the *Chlamydomonas* Resource Center (St. Paul, MN). Cells were cultured as previously described[68]. Briefly, strains were maintained on plates of TAP medium (20 mM Tris HCl, 7 mM NH$_4$Cl, 0.40 mM MgSO$_4$, 0.34 mM CaCl$_2$, 2.5 mM Na$_3$PO$_4$, and 1000 × diluted Hutner's trace elements, titrated to pH 7.0 with glacial acetic acid)[69] containing 1.5% agar. For axonemal tubulin purification, cells were resuspended in liquid TAP media and grown with continuous aeration and illumination at 21 °C for ~7 d.

**Bovine brain tubulin purification and labeling**. Bovine brain tubulin was purified by two cycles of polymerization-depolymerization and stored in 1×BRB80 (80 mM PIPES, 1 mM MgCl$_2$, 1 mM EGTA, pH 6.8)[70]. Protein concentration was determined by NanoDrop 2000 spectrophotometer (Thermo Fisher Scientific, USA). Purified protein was snap-frozen using liquid nitrogen and stored at −80 °C until use. Tetramethylrhodamine-labeled tubulin was labeled using 5(6)-TAMRA, succinimidyl ester (Invitrogen, USA) by incubating the dye with polymerized MTs. Labeled tubulin was obtained with additional cycles of polymerization and depolymerization.

**Axonemal tubulin purification**. *C. reinhardtii* cells were harvested and axonemes were isolated by standard methods. Cells were deflagellated using 4.2 mM dibucaine-HCl, and flagella were demembranated by 1% IGEPAL CA-630. Tubulin was then extracted by cycles of NaNO$_3$ (330 mM/500 mM) salt extraction solution and centrifugation at 125,000 g. Supernatant was collected and further purified by 5 ml

HiTrap Q HP column (GE, USA), followed by size exclusion Superdex 200 increase 10/300 GL column (GE, USA). The purified axonemal tubulin protein was eluted in 1×BRB80 and samples were snap-frozen using liquid nitrogen and stored at −80 °C until use. For cycled axonemal tubulin, purified tubulin was polymerized for 1 h in 1×BRB80 supplemented with 1 mM GTP, 3.5 mM MgCl$_2$ and 10% DMSO. Next, the sample was layered on cushion buffer (1×BRB80 supplemented with 60% glycerol) and centrifuged at 37 °C for 40 min at 80,000 g. The MTs were then resuspended in cold 1×BRB80 and incubated on ice for additional 20 min. The sample was centrifuged once again at 4 °C for 20 min at 80,000 × g and the supernatant was collected and snap-frozen using liquid nitrogen and stored at −80 °C until use. Protein concentration was determined by NanoDrop 2000 spectrophotometer.

**SDS-PAGE and Immunoblot**. SDS-PAGE was performed using 4–15% Mini-PROTEAN TGX precast protein gels that were stained in Coomassie blue. For Western blot analysis, the following primary antibodies were used: Gly-Pep1 anti-mono/biglclated tubulin (1:10,000; AdipoGen)[17] anti-acetylated α-tubulin (clone 6–11B-1; 1:50,000; MilliporeSigma); anti-polyglutamylation (clone GT335; 1:1000; AdipoGen); anti-detyrosinated tubulin (1:1500); anti-Δ2 tubulin (1:1000; MilliporeSigma) anti α-tubulin (clone B-5-1-2; 1:4000; MilliporeSigma) anti-tyrosinated-tubulin (1:1000; MilliporeSigma).

**Electron microscopy**. For transmission electron microscopy (TEM), axonemes were fixed (0.1 M sodium cacodylate buffer, 2.5% glutaraldehyde, 2% paraformaldehyde, pH 7.4) for 30 min at room temperature, followed by 30 min at 4 °C. Samples were then rinsed 3 times in 0.1 M sodium cacodylate buffer, trimmed and postfixed in 1% osmium tetroxide for 1 h, en bloc stained in 2% uranyl acetate in maleate buffer at pH 5.2 for a another hour, rinsed and then dehydrated in an ethanol series, infiltrated with resin (Embed812 Electron Microscopy Science), and baked overnight at 60 °C. Hardened blocks were cut using a Leica UltraCut UC7, and 60-nm sections were collected on formvar/carbon-coated nickel grids and stained using 2% uranyl acetate and lead citrate. Samples were imaged using a FEI Tecnai Biotwin transmission electron microscope. Images were taken using Morada CCD and iTEM (Olympus) software. For negative staining, 5 µl samples were deposited on a glow-discharged formvar/carbon-coated copper grid (Electron Microscopy Sciences) and stained with 2% uranyl acetate. Imaging was performed on a JEOL JEM-1400 Plus microscope.

**Microscopy assay and imaging conditions**. For dynamic MT assays, axonemal tubulins extensions were grown from TAMRA-labeled GMPCPP-stabilized bovine brain MT seeds and imaged by IRM[32]. Coverslips (Marienfeld, Germany) were cleaned using piranha solution and rendered hydrophobic by silinization[71]. Experiments were performed in a flow channel formed by two Parafilm strips between $18 \times 18$ mm$^2$ and $22 \times 22$ mm$^2$ coverslips.

MTs growth was initiated by perfusing polymerization solution BRB80 supplemented with oxygen scavengers (20 mM glucose, 20 µg/ml glucose oxidase, 8 µg/ml catalase, 0.1 mg/ml casein, 1 mM dithiothreitol and 1 mM GTP) and tubulin at different concentrations. Dynamic assays were performed at 28 °C and imaged at frame rates of 0.2 fps for growth rate analysis and 10 fps for shrinkage rate analysis.

For TIRF microscopy imaging, axonemal MTs were mixed with 488-Alexa labeled bovine brain tubulin (final concentration of 8% bovine brain tubulin) and imaged similarly as the IRM imaging.

**Identification of plus and minus ends**. Because the growth rates at the plus and minus ends of axonemal MTs are sometimes indistinguishable, we used polarity marked MT seeds to identify the ends of the MTs. On average, the plus ends grew faster than the minus ends (Supplementary Fig. 13), as expected. Furthermore, 76 out of 85 of these polarity marked MTs grew faster at their plus ends than their minus. Therefore, in the IRM measurements, we defined the MTs plus end as the end that grew more. The error in the assignment of MT polarity is therefore approximately 10%.

**Dual-labeled GMPCPP stabilized MT seeds**. GMPCPP stabilized seeds were polymerized from 2 µM TAMRA-labeled bovine brain tubulin for 20 min at 37 °C. Then, the MTs were centrifuged for 5 min at 20 psi in an Airfuge, and resuspended in warm polymerization solution (BRB80, 1 mM GMPCPP, 1 mM MgCl$_2$, 0.6 µM Alexa488-labeled bovine brain tubulin). The solution was incubated for 30 min at 37 °C, and then centrifuged for 5 mins in an Airfuge and resuspended in 200 µL warm BRB80.

**Image analysis**. Image analysis was performed by creating kymographs of individual MT seeds using ImageJ analysis software (National Institutes of Health). Growth rates were calculated from the slopes of the kymographs.

**Statistics**. The data plotting and curve fitting were performed with Prism 7 (Graph-Pad). Evaluations of statistical significance are described in the respective figure legends. Biological replicates are measurements of tubulin from different

**Table 4 Tubulin sequences that were used in this study**

| Organism | Protein name | UniPort Accession No | Gene |
|---|---|---|---|
| C. reinhardtii | Tubulin beta-1/beta-2 | P04690 | TUB1 |
| P. tetraurelia | Tubulin beta chain | P33188 | bPT2 |
| T. thermophila | Tubulin beta chain | P41352 | BTU1 |
| V. carteri | Tubulin beta chain | P11482 | TUBB1 |
| S. scrofa | Tubulin beta chain | Q767L7 | TUBB2B |
| S. cerevisiae | Tubulin beta chain | P02557 | TUB2 |
| D. melanogaster | Tubulin beta-3 | P08841 | Tub60D |
| B. taurus | Tubulin beta-2B | Q6B856 | TUBB2B |
| | Tubulin beta-3 | Q2T9S0 | TUBB3 |
| | Tubulin beta-4A | Q3ZBU7 | TUBB4A |
| | Tubulin beta-4B | Q3MHM5 | TUBB4B |
| | Tubulin beta-5 chain | Q2KJD0 | TUBB5 |
| | Tubulin beta-6 | Q2HJ81 | TUBB6 |
| H. sapiens | Tubulin beta-1 | Q9H4B7 | TUBB1 |
| | Tubulin beta-2A | Q13885 | TUBB2A |
| | Tubulin beta-2B | Q9BVA1 | TUBB2B |
| | Tubulin beta-3 | Q13509 | TUBB3 |
| | Tubulin beta-4A | P04350 | TUBB4A |
| | Tubulin beta-4B | P68371 | TUBB4B |
| | Tubulin beta chain | P07437 | TUBB |
| | Tubulin beta 8 | Q3ZCM7 | TUBB8 |

purifications. Technical replicates are measurements of tubulin from the same purification that were done on different days. All experiments were performed at least three times.

**Sequences**. Table 4 presents the sequences that were used in this study. Tubulin sequences have been aligned with Geneious software. Amino acid homologies are marked in dots.

**MSD analysis**. To determine the diffusion coefficient and net growth rate of axonemal tubulin, the end position of the MTs was tracked using FIESTA[72]. Data was then analyzed using the MSD analyzer package[73], and values were calculated by quadratic fitting to the first 15 s.

**PFs analysis**. The lengths of the MT tips were measured manually using ImageJ software. Start point of the measurement was determined by reduced intensity of the MT. Curvature measurements were performed in MATLAB by fitting a circle to the last 0.6 µm of the tip.

**Reporting summary**. Further information on research design is available in the Nature Research Reporting Summary linked to this article.

## Data availability
The data supporting the findings of this manuscript are available from the corresponding author upon reasonable request. A reporting summary for this Article is available as a Supplementary Information file. The source data underlying Figs. 1a, b, d, 2c, e, g–i, 3, and Supplementary Figs 1, 4, 5–7 are provided as a Source Data file.

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

## Acknowledgements

We thank Dr. Xinran Liu from the core center for cellular and molecular imaging at Yale for sample processing, sectioning and image acquisition, and Dr. Joel Nott from the protein facility at Iowa State University for the mass-spectrometry analysis. Special thanks to Dr. Sabyasachi Sutradhar for the great help with MATLAB and all The Howard lab members for fruitful discussion. Research reported in this publication was supported by a Cross-Disciplinary Fellowship from Human Frontier Science Program (LT000919/2015-C) and European Molecular Biology Organization long-term fellowship (ALTF 1424-2014) to RO, and National Institute of General Medicine Sciences of the National Institutes of Health under award number R01GM110386 to JH.

## Author contributions

R.O. designed and performed all experiments. R.O and J.H. analyzed the data and wrote the paper.

## Additional information

**Competing interests:** The authors declare no competing interests.

