## [Peer Review File · Nature Communications]

Reviewers' Comments:

Reviewer #1:

Remarks to the Author:

In their manuscript, Orbach and Howard develop a meticulous method to isolate different, defined fractions of axonemal tubulin and analyse the biochemical and biophysical properties of each of these tubulin fractions. Employing controlled purification and extraction conditions allowed them to obtain strongly enriched fractions of tubulin from 1) the central pair, 2) the B-tubules, and 3) and A-tubules of *Chlamydomonas* axonemes. This is a huge step forward in the current effort to further deciphering the functional roles of tubulin diversity, referred to as the tubulin code. While recently many teams have successfully purified tubulin from different sources, this study is to my knowledge the first to purify distinct fractions of tubulin from one and the same source – the flagellum. The rigorous purification of the three axonemal tubulin fractions plus their exhaustive characterization alone will make a strong impact on the field. Moreover, the authors discover a novel growth behaviour of microtubules, which is most likely directly dependent on the tubulin composition of axonemes: they show that during growth, plus ends become thinner and then curl away from the microtubule lattice. This behaviour has never been reported for brain tubulin and might therefore be a specific property of axonemal tubulin. This discovery also provides further evidence for the intrinsic curvature of the GTP-tubulin dimer, and allows the authors to propose an axoneme-specific model of microtubule growth.

Overall, the manuscript is well-written, logically structured, experiments are carefully reported and figures are of high quality, which makes reading and understanding easy. Despite the high quality of the current manuscript, there are some points the authors need to consider.

Major points:

1) The whole point of this paper is to show that axonemal tubulin has different properties than brain tubulin. Brain tubulin has been the most-used tubulin source for all kind of biophysical or biochemical experiments performed over the past 50 years, and many so far generalized assumptions of tubulin behaviour have been solely drawn from experiments with brain tubulin. In this sense, it is extremely important to show that some of the “canonical” tubulin and microtubule properties might be not true for all types of tubulin, hence the importance of the current work. However, to be able to draw a convincing conclusion, the authors should prepare brain tubulin using at least some of the steps they use in the purification of axonemal tubulin, for instance, the salt extraction and the purification on an affinity column. Otherwise, the doubt that the different behaviour of microtubules polymerised from brain or axonemal tubulin are purification-related artefacts might remain, thus lowering the impact of this work. Alternatively, the authors could consider to pass the axonemal tubulin through cycles of polymerization and depolymerization to show that its specific properties remain even after this procedure, which is used to purify brain tubulin.

This concern becomes particularly important for the last observation of the authors: the curved, thinning growth of microtubule tips. One could imagine that this phenomenon happens because in contrast to brain tubulin, axonemal tubulin has not been purified by repeated cycles of polymerization and depolymerization. Therefore, polymerisation-incompetent tubulin dimers could be present, which could, upon local accumulation at growing microtubule ends, partially interrupt the synchronous assembly of all protofilaments, thus leading to the elongation of only some protofilaments. Those could then, as a result of the missing lattice interactions, bend outward. This point must absolutely be clarified to solidify this exciting observation.

2) The authors characterise the PTMs of tubulin on different fractions of axonemal tubulin, and confirm that polyglutamylation is enriched on B-tubules as compared to A-tubules. This is an important proof of a concept that was previously proposed based on immunoelectron microscopy. The authors also analyse many of the other known PTMs of tubulin, however in contrast to polyglutamylation, they do not find differences in the distribution of these PTMs between different fractions (Fig. 3a).

As there is no intrinsic difference, it is very hard to conclude how much of these different PTMs are present on the tubulin analysed. It could be everything from very low to very high coverage, as there is no comparison to a tubulin species with known levels of those PTMs.

The authors might want to verify this by including well-characterized brain tubulin and if available mammalian axonemal tubulin (sperm tails) and unmodified mammalian tubulin in their analyses in order to have some reference points for their observations. Otherwise, only the analysis of polyglutamylation could be considered fully conclusive.

(For readability, these important western blots should be shown as a bigger panel, and the blots for the different PTMs should be less cropped.)

3) Discussion page 19: role of tubulin isotypes. The authors search for specific features of *Chlamydomonas* tubulin that could explain the specific behaviour of the purified axonemal tubulin (supplementary Fig. S4, S5, S6, S7). However, they do not discuss the fact that the tubulin they isolated from *Chlamy* flagella is the same throughout the *Chlamy* cells, as there is only one single isotype for alpha- and beta-tubulin. So does this mean that cytosolic or spindle microtubules in *Chlamy* are also less dynamic? In other words, does the tubulin isotype from *Chlamy* really explain cilia-specific feature of tubulin?

Minor points:

1) In the abstract, the authors write: "Together, our study provides new insights into growth, stability and the role of post-translational modifications of microtubules."

This is an overstatement, they rather show this for axonemal microtubules. This should be clearly stated.

2) Page 3 line 14: following the description of the stable axonemal microtubules, the authors state that microtubules are highly dynamic. For an outsider of the field this might sound like a contradiction. They might want to specify that they talk about non-axonemal microtubules.

3) Page 4 line 3-4: the authors write that in this paper they address the role of tubulin posttranslational modifications in controlling the dynamic properties of axonemal microtubules. However, what they really do is to compare two different tubulin fractions which appear to have different levels of tubulin polyglutamylation, but they have only addressed the role of this PTM in suppl. Fig. S2. Why is this important result not part of the main manuscript?

4) The introduction is missing a part about the tubulin code, the existence of different tubulin isotypes and PTMs. As this is the whole point of the current paper, this should be shifted from the discussion to the introduction. Also, page 4 lines 10-14 should be moved into the introduction, just after the explanation of the tubulin code. This would make the rationale of the whole work much clearer to the reader.

5) To better introduce the key advance of this work; the purification of different axonemal tubulin fractions, the authors might want to give a brief introduction into the difficulties past studies have encountered in purifying axonemal tubulin. This could be a great starting point for the result section (again, perhaps move elements from the discussion).

6) Throughout the text, the authors should make sure to use the term "bovine brain tubulin" and not "bovine tubulin".

7) Page 10 line 13: there is a problem with this sentence.

8) Fig. 3c: the nomenclature of the western blot is not clear, what does WT refer to? Is it the entire axoneme?

9) Page 17 lines 12-14: the authors discuss the possible involvement of detyrosination in the

axoneme, and compare it to the muscle microtubules for which detyrosination had been shown to play a role in mechanotransduction. However, the proposed model for the muscle also involves interactions with other muscle elements, such as intermediate desmin filaments. This is clearly different from the cilia, and should be discussed a bit more in detail here.

10) Discussion page 18 should be re-considered in the light of Major point 1).

11) Page 18 lines 12-16: could be authors develop why they think the observations at high tubulin concentrations are also valid at low concentrations despite the fact that they did not see it? The current explanations seem not fully comprehensible.

12) Supplemental Fig. S5, S7: in c, how did the authors choose the tubulin isotypes from different organisms to compare? Are these ciliary tubulin isotypes? Could they explain this, and also annotate these isotypes in the figure?

13) Discussion page 20 lines 4-12: the authors write that the canonical model of tubulin polymerisation is that GTP-tubulin dimers are straight, while GDP-dimers are curved. They should tone this down, as it has long been accepted that this is not the case, and both dimers are curved.

14) Supplementary movies: It might help to show some frames of each movie in the main figure they relate to, in order to facilitate the rapid reading of the paper.

Reviewer #2:

Remarks to the Author:

The dynamic and structural properties of axonemal tubulins support the high length stability of tubulins
Orbach and Howard

The microtubule field is currently experiencing a renaissance in our appreciation of the molecular diversity of tubulin proteins and how this diversity contributes to microtubule function. This timely study by Orbach and Howard addresses a long-standing question in the field – are the unique microtubule architectures within axonemes determined by tubulins with unique biochemical activities? The strengths of this study are the importance of the topic, and the combination of elegant biochemical approaches to isolate different populations of axonemal tubulins and compare their dynamic assembly behaviors and structures. The results shed new light on the molecular properties of axonemal tubulins, and could be extended to more broadly improve our understanding of microtubule structure and function.

In its current form, however, the manuscript has several major flaws that limit the impact of the findings. These issues, which are detailed below, should be addressed by the authors to strengthen manuscript, at which point it could be suitable for publication in Nature Communications. An overarching concern is whether the observed differences in microtubule dynamics and structure are attributable to differences between axonemal tubulin vs brain tubulin, or rather Chlamydomonas tubulin vs mammalian brain tubulin. The results clearly identify some interesting differences between tubulins, but these differences do not seem to be determined by axoneme-specific PTMs or MAPs, or by tubulin in the CP and B-tubule fractions. So perhaps the results are revealing general differences between tubulins from different species, rather than axoneme-specific differences?

Major Points:

1) The impact of the study critically depends on the purity of the ciliary tubulin fractions. Presumably, there must be some distinct biochemical components in these fractions that

determine their salt sensitivity in the fractionation procedure. Some protein other than tubulin would be a reasonable candidate for these effects; therefore, it is important for the authors to rigorously test for the presence of other proteins in the relevant fractions. The mass spec data in the supplemental material is a good start, but it is not clear which fraction this data corresponds to – CP, B-tubule, or A-tubule? A simple, alternative approach would be to run samples of the fractions used in dynamics and EM experiments on SDS PAGE at high protein loads, and stain by coomassie to enable the detection of trace proteins. This point must be addressed to strengthen the conclusion that the observed effects are attributable to tubulins, rather than other components of the ciliary fractions.

2) It is not clear whether the microtubule dynamics experiments with ciliary tubulins were conducted at a different temperature than the experiments with bovine brain tubulin. This is an important point that must be clarified in order to interpret the dynamics data, because temperature differences are expected to have strong effects on microtubule dynamics. The Materials and Methods section (p29, line 13) states that experiments were conducted at 28°C. I suspect this was used for the ciliary tubulins, but not the brain tubulin, which should not assemble at this temperature at the given concentration. If the bovine brain tubulin reactions were conducted at a different temperature, the authors must make this clear, and also clearly describe how they accounted for this difference so that they could accurately compare data from these different experiments.

3) Figure 2: it is not clear what tubulin fraction was used for these experiments. The first line on page 7 states that “a mixture of CP and B-tubule tubulins” were used. What fraction from Figure 1 does this correspond to? I am concerned that this fraction may contain contaminating MAPs that could alter dynamics. It is therefore important for the authors to identify the tubulin sample and demonstrate its purity by either SDS PAGE or mass spec.

On a related note, it would be useful for the authors to state whether the observed depolymerization rate (Figure 2h) exhibits concentration dependence. One would not expect this, based on previous work from the authors' lab and others; however, if there is a MAP in the protein sample that impacts depolymerization, then this MAP might exhibit concentration-dependent effects. Therefore, it would be informative for the authors to test for concentration dependence of depolymerization.

4) Figure 3: the comparison of tubulin posttranslational modifications in the different tubulin fractions is a potentially exciting advance for the field. These results would be strengthened by adding quantifications of band intensities, plots comparing these intensities between fractions, and demonstration of reproducibility across multiple experiments.

5) Page 12, line 17-18: The conclusion that “In both strains no significant differences were observed in their dynamic properties” is not supported by the data in Figure 3 and Supplemental Figure 2. Supplemental Figure 2 shows depolymerization rates that are ~100X slower than those in Figure 3e, and Catastrophe frequencies that are ~50% lower.

6) Axonemal MTs form curved tip structures (p13-16). It is not clear which tubulin fraction was used for this analysis. Are similar characteristics observed across all fractions; CP, B-tubule and A-tubule? This must be clarified.

I'm concerned about the interpretation of MT signal anomalies in the IRM experiment. Specifically, I worry about interpreting the differences in signal as differences in pf number. For example, could curled segments at the plus ends be MTs bending out of the imaging plane rather than protofilaments? Since this imaging technique is rather new to the field, the impact of these findings would be strengthened by repeating these morphological studies using the more conventional TIRF method, with a low concentration of added labeled tubulin, and testing for similar evidence of curled protofilaments and partial lattices.

Also in this section, it would be useful to include data from experiments with bovine brain tubulin as a control. This would help mitigate concern about potential experimental artifacts.

7) Overall, the discussion is quite interesting and well written. One point that the authors should consider including is their finding that axonemal tubulin depolymerizes faster than brain tubulin (Figure 2h). I find it quite interesting that the ciliary tubulin assembles slower and catastrophes less often, but appears to depolymerize 1.5X faster. This seems inconsistent with the conclusion that axonemal tubulin has "high length stability" (Discussion, p16, line 18-19).

Minor Points:

1) Typo on p10, line 13: "Thus, we could soluble..."

2) Page 13, line 7: the units used to indicate curvature (μm^{-1}) seem insufficient. Perhaps this should be a degree of curvature per micron? Or a radius of curvature?

3) Typo on p18, line 11: "maybe" should read "may be"

4) p19, second paragraph: The numbering of the amino acid positions appears to be incorrect. The Lys residue in α -tubulin is at position 58, not 60. The Tyr residue is at position 222 in α -tubulin, not 224. The correct positions are shown in the alignment in supplemental figure

Reviewers' comments:

Reviewer #1 (Remarks to the Author):

Major points:

1) The whole point of this paper is to show that axonemal tubulin has different properties than brain tubulin. Brain tubulin has been the most-used tubulin source for all kind of biophysical or biochemical experiments performed over the past 50 years, and many so far generalized assumptions of tubulin behaviour have been solely drawn from experiments with brain tubulin. In this sense, it is extremely important to show that some of the “canonical” tubulin and microtubule properties might be not true for all types of tubulin, hence the importance of the current work. However, to be able to draw a convincing conclusion, the authors should prepare brain tubulin using at least some of the steps they use in the purification of axonemal tubulin, for instance, the salt extraction and the purification on an affinity column. Otherwise, the doubt that the different behaviour of microtubules polymerised from brain or axonemal tubulin are purification-related artefacts might remain, thus lowering the impact of this work. Alternatively, the authors could consider to pass the axonemal tubulin through cycles of polymerization and depolymerization to show that its specific properties remain even after this procedure, which is used to purify brain tubulin.

We do not believe that the differences between the *Chlamydomonas* axonemal tubulins and bovine brain tubulin are due to exposure to salt. At the start of the work with the different salt extractions, we tested the ability of bovine tubulin to withstand 500mM NaNO₃ exposure (i.e. to remain polymerization competent). In response to the reviewer's comments, we have examined the effect of this salt on bovine brain tubulin in more detail. Bovine brain tubulin was incubated with 500mM NaNO₃ for 2 hours and followed by size exclusion chromatography to allow buffer exchange. No curved tip structures were observed in subsequent assays. The dynamic properties were slightly different, namely both the growth rate and the catastrophe frequency were somewhat higher, though the increase was not statistically significant. These results actually strengthen our conclusion that axonemal tubulin is less dynamic than bovine tubulin because we found that axonemal tubulin had a lower growth rate and catastrophe frequency (i.e. the salt had a slight affect in the opposite direction). These results are shown in Supplementary Figure 1.

See below for results using cycled axonemal tubulin: the tip structures were still observed.

This concern becomes particularly important for the last observation of the authors: the curved, thinning growth of microtubule tips. One could imagine that this phenomenon happens because in contrast to brain tubulin, axonemal tubulin has not been purified by repeated cycles of polymerization and depolymerization. Therefore, polymerisation-

incompetent tubulin dimers could be present, which could, upon local accumulation at growing microtubule ends, partially interrupt the synchronous assembly of all protofilaments, thus leading to the elongation of only some protofilaments. Those could then, as a result of the missing lattice interactions, bend outward. This point must absolutely be clarified to solidify this exciting observation.

Following the reviewer's suggestion, we tested axonemal tubulin that was cycled after purification. We observed similar tip structures after cycling, as before cycling. Videos of this result is included in the supplementary material and is referred to in the text.

2) The authors characterise the PTMs of tubulin on different fractions of axonemal tubulin, and confirm that polyglutamylation is enriched on B-tubules as compared to A-tubules. This is an important proof of a concept that was previously proposed based on immunoelectron microscopy. The authors also analyse many of the other known PTMs of tubulin, however in contrast to polyglutamylation, they do not find differences in the distribution of these PTMs between different fractions (Fig. 3a). As there is no intrinsic difference, it is very hard to conclude how much of these different PTMs are present on the tubulin analysed. It could be everything from very low to very high coverage, as there is no comparison to a tubulin species with known levels of those PTMs.

In response to reviewer #2, we quantified the band intensities of immunoblots derived from the three independent purifications in Supplementary Fig. 3. We found less immunoreactivity in the A-tubule fraction for detyrosination, $\Delta 2$ -tubulin and mono/biglycylation. Thus, we believe that the CP and B-tubule signals for these PTMs are real (i.e. not background). Note that the quantification supports our original conclusions that there is a marked increase in polyglutamylation in the B-tubule (compared to the central pair and the A-tubule).

The authors might want to verify this by including well-characterized brain tubulin and if available mammalian axonemal tubulin (sperm tails) and unmodified mammalian tubulin in their analyses in order to have some reference points for their observations. Otherwise, only the analysis of polyglutamylation could be considered fully conclusive.

For acetylation, we used yeast tubulin as a control. Earlier experiments by Alper et al. (2013) using enzymatic deacetylation also confirmed that the acetylation is real.

(For readability, these important western blots should be shown as a bigger panel, and the blots for the different PTMs should be less cropped.)

We have redesigned this panel to improve the readability.

3) Discussion page 19: role of tubulin isotypes. The authors search for specific features of Chlamydomonas tubulin that could explain the specific behaviour of the purified axonemal tubulin (supplementary Fig. S4, S5, S6, S7). However, they do not

discuss the fact that the tubulin they isolated from Chlamy flagella is the same throughout the Chlamy cells, as there is only one single isotype for alpha- and beta-tubulin. So does this mean that cytosolic or spindle microtubules in Chlamy are also less dynamic? In other words, does the tubulin isotype from Chlamy really explain cilia-specific feature of tubulin?

This is a very important point, which was also raised by the second reviewer. We have elaborated on this point in the discussion. While the cytosolic microtubules in Chlamy are dynamic (which may be due to MAPs), there are sequence differences between ciliate and non-ciliate tubulins that may lead to more stability in the former. On the other hand, there are sequences specific to metazoan ciliary tubulins that are also common to Chlamy and other ciliate tubulins, suggesting that low dynamicity is a property of axonemal tubulin. This question is still open.

Minor points:

1) In the abstract, the authors write: "Together, our study provides new insights into growth, stability and the role of post-translational modifications of microtubules." This is an overstatement, they rather show this for axonemal microtubules. This should be clearly stated.

We changed that last sentence of the abstract from microtubules to axonemal microtubules.

2) Page 3 line 14: following the description of the stable axonemal microtubules, the authors state that microtubules are highly dynamic. For an outsider of the field this might sound like a contradiction. They might want to specify that they talk about non-axonemal microtubules.

As suggested, we added non-axonemal tubulin to this paragraph.

3) Page 4 line 3-4: the authors write that in this paper they address the role of tubulin posttranslational modifications in controlling the dynamic properties of axonemal microtubules. However, what they really do is to compare two different tubulin fractions which appear to have different levels of tubulin polyglutamylation, but they have only addressed the role of this PTM in suppl. Fig. S2. Why is this important result not part of the main manuscript?

We moved Fig. S2 to the main text (Fig. 3).

4) The introduction is missing a part about the tubulin code, the existence of different tubulin isoforms and PTMs. As this is the whole point of the current paper, this should be shifted from the discussion to the introduction. Also, page 4 lines 10-14

should be moved into the introduction, just after the explanation of the tubulin code. This would make the rationale of the whole work much clearer to the reader.

We added a short introduction to PTMs and moved the first paragraph from the results to the introduction.

5) To better introduce the key advance of this work; the purification of different axonemal tubulin fractions, the authors might want to give a brief introduction into the difficulties past studies have encountered in purifying axonemal tubulin. This could be a great starting point for the result section (again, perhaps move elements from the discussion).

A brief introduction discussing the difficulties of previous studies to purify functional axonemal tubulin was added to the last paragraph of the introduction.

6) Throughout the text, the authors should make sure to use the term “bovine brain tubulin” and not “bovine tubulin”.

We changed the text from bovine tubulin to bovine brain tubulin.

7) Page 10 line 13: there is a problem with this sentence.

We rephrased the sentence.

8) Fig. 3c: the nomenclature of the western blot is not clear, what does WT refer to? Is it the entire axoneme?

“WT” is not the right term. We changed it to “CP/B”, corresponding to tubulins purified from the combined Cyc1-Cyc5 extracts (i.e. tubulins from the central pair and the B-tubule). We added an explanation in the text and in the figure legend to clarify this point.

9) Page 17 lines 12-14: the authors discuss the possible involvement of detyrosination in the axoneme, and compare it to the muscle microtubules for which detyrosination had been shown to play a role in mechanotransduction. However, the proposed model for the muscle also involves interactions with other muscle elements, such as intermediate desmin filaments. This is clearly different from the cilia, and should be discussed a bit more in detail here.

We decided to remove this part from the paper since it is hypothesis.

10) Discussion page 18 should be re-considered in the light of Major point 1).

As mentioned in point 1, to confirm our observation of curved tip structures, we performed two sets of experiments: (i) cycling axonemal tubulin. (ii) examining the

effect of NaNO₃ on bovine microtubules. Our conclusions remained the same – axonemal microtubules form curved tip structures, while bovine tubulin do not show such structures.

11) Page 18 lines 12-16: could be authors develop why they think the observations at high tubulin concentrations are also valid at low concentrations despite the fact that they did not see it? The current explanations seem not fully comprehensible.

We reworded this paragraph to make it clearer:

“Tip structures were detected primarily at high concentrations. However, some tip structures were also seen at low tubulin concentration. The reason that fewer tip structures were seen at lower concentration could be that the tip structures are shorter due to the lower growth rates. Thus, it is possible that the tip structures are more prevalent than we reported. If this is the case, then tip structures could contribute to the high growth variability of the axonemal MTs as revealed by our MSD analysis.”

12) Supplemental Fig. S5, S7: in c, how did the authors choose the tubulin isotypes from different organisms to compare? Are these ciliary tubulin isotypes? Could they explain this, and also annotate these isotypes in the figure?

We annotated the tubulin isotypes in both figures.

All tubulin isotypes of non-ciliates organisms do not have Arg-60 (most isotypes have Lys and sometimes Asp). In contrast, all tubulin isotypes of ciliates have Arg-60. Regarding Tyr-224, Only *Chlamydomonas* and *Volvox* have Phe-224. We clarified these points in the text.

13) Discussion page 20 lines 4-12: the authors write that the canonical model of tubulin polymerisation is that GTP-tubulin dimers are straight, while GDP-dimers are curved. They should tone this down, as it has long been accepted that this is not the case, and both dimers are curved.

Following the reviewer’s suggestion, we rephrased the paragraph and referred to the “straighter conformation” of GTP-tubulin.

14) Supplementary movies: It might help to show some frames of each movie in the main figure they relate to, in order to facilitate the rapid reading of the paper.

Images from different movies were added to Figure 4.

Reviewer #2 (Remarks to the Author):

In its current form, however, the manuscript has several major flaws that limit the impact of the findings. These issues, which are detailed below, should be addressed by the authors to strengthen manuscript, at which point it could be suitable for publication in

Nature Communications. An overarching concern is whether the observed differences in microtubule dynamics and structure are attributable to differences between axonemal tubulin vs brain tubulin, or rather Chlamydomonas tubulin vs mammalian brain tubulin. The results clearly identify some interesting differences between tubulins, but these differences do not seem to be determined by axoneme-specific PTMs or MAPs, or by tubulin in the CP and B-tubule fractions. So perhaps the results are revealing general differences between tubulins from different species, rather than axoneme-specific differences?

The reviewer is quite correct in that the differences may be related to different species rather than to axonemal vs. cytoplasmic tubulins. This point was also raised by Reviewer #1. Please see our response to their major point (3) above. We have now made our arguments clearer in the text.

Major Points:

1) The impact of the study critically depends on the purity of the ciliary tubulin fractions. Presumably, there must be some distinct biochemical components in these fractions that determine their salt sensitivity in the fractionation procedure. Some protein other than tubulin would be a reasonable candidate for these effects; therefore, it is important for the authors to rigorously test for the presence of other proteins in the relevant fractions. The mass spec data in the supplemental material is a good start, but it is not clear which fraction this data corresponds to – CP, B-tubule, or A-tubule? A simple, alternative approach would be to run samples of the fractions used in dynamics and EM experiments on SDS PAGE at high protein loads, and stain by coomassie to enable the detection of trace proteins. This point must be addressed to strengthen the conclusion that the observed effects are attributable to tubulins, rather than other components of the ciliary fractions.

SDS-PAGE with high tubulin load (15 ug) showed that both CP and B-tubule fractions include only low molecular weight molecules, which are associated with tubulin degradation. An image of the SDS-PAGE gel was added (Supplementary Fig. 6). Furthermore, following reviewer #1 comments we cycled axonemal tubulin after purification and observed similar tip structures.

2) It is not clear whether the microtubule dynamics experiments with ciliary tubulins were conducted at a different temperature than the experiments with bovine brain tubulin. This is an important point that must be clarified in order to interpret the dynamics data, because temperature differences are expected to have strong effects on microtubule dynamics. The Materials and Methods section (p29, line 13) states that experiments were conducted at 28°C. I suspect this was used for the ciliary tubulins, but not the brain tubulin, which should not assemble at this temperature at the given concentration. If the bovine brain tubulin reactions were conducted at a different temperature, the authors must make this clear, and also clearly describe how they

accounted for this difference so that they could accurately compare data from these different experiments.

All the reported experiments were performed at the same temperature of 28°C. In our hands, bovine brain tubulin does not polymerize at 26°C, but at 28°C we had no problem to polymerize MTs.

3) Figure 2: it is not clear what tubulin fraction was used for these experiments. The first line on page 7 states that “a mixture of CP and B-tubule tubulins” were used. What fraction from Figure 1 does this correspond to?

In these experiments we used tubulin that was purified from the pooled fractions Cyc 1-Cyc 5 (SE fraction from Fig. 1d) (labeled “CP/B”). We now clarify this point in the text.

I am concerned that this fraction may contain contaminating MAPs that could alter dynamics. It is therefore important for the authors to identify the tubulin sample and demonstrate its purity by either SDS PAGE or mass spec.

The tubulin from these fractions was purified using ion-exchange and size exclusion chromatography with high purity as judged by SDS-PAGE and mass spec (see point 1) above.

On a related note, it would be useful for the authors to state whether the observed depolymerization rate (Figure 2h) exhibits concentration dependence. One would not expect this, based on previous work from the authors’ lab and others; however, if there is a MAP in the protein sample that impacts depolymerization, then this MAP might exhibit concentration-dependent effects. Therefore, it would be informative for the authors to test for concentration dependence of depolymerization.

We measured the depolymerization rate of the microtubules at different tubulin concentrations and found no difference (Supplementary Figure 7). The lack of concentration dependence argues against MAPs influencing depolymerization. We thank the reviewer for making this point.

4) Figure 3: the comparison of tubulin posttranslational modifications in the different tubulin fractions is a potentially exciting advance for the field. These results would be strengthened by adding quantifications of band intensities, plots comparing these intensities between fractions, and demonstration of reproducibility across multiple experiments.

We quantified the band intensities as suggested (Supplementary Fig. 3). The quantifications of the three independent tubulin purifications show that the results are reproducible and we have referred to Fig. S3 in the text. The quantification also allowed us to answer point 2 raised by Reviewer #1. We thank the reviewer for this suggestion.

5) Page 12, line 17-18: The conclusion that “In both strains no significant differences were observed in their dynamic properties” is not supported by the data in Figure 3 and Supplemental Figure 2. Supplemental Figure 2 shows depolymerization rates that are ~100X slower than those in Figure 3e, and Catastrophe frequencies that are ~50% lower.

Regarding shrinkage rate, the label in Supplemental Figure 2 was wrong and has been corrected to growth rate (this is now Figure 3h). Regarding catastrophe frequencies in Figure 3, the differences are less than 50% and not statistically significant between the triplicates.

6) Axonemal MTs form curved tip structures (p13-16). It is not clear which tubulin fraction was used for this analysis. Are similar characteristics observed across all fractions; CP, B-tubule and A-tubule? This must be clarified.

All tubulin fractions (CP, B-tubule, and their mixture CP/B) showed curved tip structures. A sentence to clarify this point was added.

I'm concerned about the interpretation of MT signal anomalies in the IRM experiment. Specifically, I worry about interpreting the differences in signal as differences in pf number. For example, could curled segments at the plus ends be MTs bending out of the imaging plane rather than protofilaments? Since this imaging technique is rather new to the field, the impact of these findings would be strengthened by repeating these morphological studies using the more conventional TIRF method, with a low concentration of added labeled tubulin, and testing for similar evidence of curled protofilaments and partial lattices.

TIRF microscopy movies (Supplementary videos 8 and 9) have included and show the same phenomena

Also in this section, it would be useful to include data from experiments with bovine brain tubulin as a control. This would help mitigate concern about potential experimental artifacts.

We added a new video that shows the growth of bovine tubulin (Supplementary video 2).

7) Overall, the discussion is quite interesting and well written. One point that the authors should consider including is their finding that axonemal tubulin depolymerizes faster than brain tubulin (Figure 2h).

The differences between the shrinkage rates are not statistically significant, which is now indicated in Figure 2h.

I find it quite interesting that the ciliary tubulin assembles slower and catastrophes less often, but appears to depolymerize 1.5X faster. This seems inconsistent with the conclusion that axonemal tubulin has “high length stability” (Discussion, p16, line 18-19).

The shrinkage rates are not statistically significant and, furthermore, the shrinkage time is almost negligible, compared to the time of growth, and therefore has almost no effect on dynamicity as defined by Toso (reference 36).

Minor Points:

1) Typo on p10, line 13: “Thus, we could soluble...”

We rephrased the sentence.

2) Page 13, line 7: the units used to indicate curvature (μm^{-1}) seem insufficient. Perhaps this should be a degree of curvature per micron? Or a radius of curvature?

To clarify the units of curvature for the reader, we added to the text explanation (curvature = 1/radius of curvature).

3) Typo on p18, line 11: “maybe” should read “may be”

We changed to “may be”.

4) p19, second paragraph: The numbering of the amino acid positions appears to be incorrect. The Lys residue in α -tubulin is at position 58, not 60. The Tyr residue is at position 222 in α -tubulin, not 224. The correct positions are shown in the alignment in supplemental figure

We corrected the numbering of the amino acids.

Reviewers' Comments:

Reviewer #1:

Remarks to the Author:

The authors have done a great effort to answer all referees' comments. They addressed all of the concerns, and also explained why certain points could not be addressed as suggested by the referee. The paper has now strongly improved. Before being accepted for publication, some rather minor points should be addressed:

1) The introduction ends with a summary of the previously used methods to extract axonemal tubulin and explaining their problems, but there is a concluding one or two sentence missing that places the current work in this context.

2) Figure 3: the figure is now much better, however the labelling of the lanes in panel b is missing. And as the figure does not fill the full page (and could thus be expanded), I would strongly suggest to put the antibodies and the PTMs detected next to each blot to facilitate the reading of this figure. It is very hard to go forth and back between legend and figure to find out which blot shows which PTM.

The Coomassie in panel a looks like a western blot – if the authors have a coloured picket of it, they should use this instead. It should also be stated in the legend or in the figure that this is a Coomassie-stained gel.

Finally, the diagrams in panels e-j have too small polices, could this be improved (in necessary, increase the overall size of these panels).

3) Figure 4 has improved. The polices in most panels are VERY small, and should be increased.

4) The authors have done an effort to replace 'bovine tubulin' with 'bovine brain tubulin', however, there are still some places where 'bovine tubulin' is used (I detected it in the Discussion). This should be corrected.

5) The authors have done a great effort to be more precise about the tubulin PTMs. I think the way they addressed the issue is satisfying. Still, there is a point that they should somehow discuss, as this is one of the most difficult technical issues in the field, and leads persistently to misleading interpretations of results.

In the discussion, they write that detyr- and $\Delta 2$ -tubulin are differently present in different axonemal fractions, while tyr-tubulin is unaltered. This is technically impossible, because alpha-tubulin can be either detyr-, $\Delta 2$ - or tyr-tubulin. Thus, if detyr- and $\Delta 2$ -tubulin change, tyr-tubulin MUST also change.

Given the results presented in Fig. 3b, I would say that detyr- and $\Delta 2$ -tubulin are clearly reduced in the pellet fraction (A-tubules?), but tyr-tubulin is not. The fact that the authors cannot MEASURE an increase in tyr-tubulin relative to the total-tubulin labelling might have other reasons: some differences are very hard to see in western blot, especially, if antibodies are extremely efficient, such as the tyr-tubulin antibody. These technical issues have been extensively discussed in a method paper (Magiera & Janke 2013), and the authors must point this out to the reader to avoid further misinterpretations of their results.

6) The authors might discuss the possibility that the substitution of Lys 58 to Arg has an effect on the potential acetylation of this residue. A previous whole-proteome study found Lys 58 acetylated (though this has never been confirmed by a study directly focusing on tubulin). Choudhary C, Kumar C, Gnad F, Nielsen ML, Rehman M, Walther TC, Olsen JV, Mann M (2009) Lysine acetylation targets protein complexes and co-regulates major cellular functions. Science 325: 834-840. It would be also interesting to think about the possibility that a substitution from Tyr to Phe has an effect on the potential of phosphorylating Tyr, but not Phe (in analogy with the acetylation).

7) The discussion about the potential role of specific sequence motifs within the C-terminal tails of tubulin should definitely be amended by the notion that not only these primary motifs, but also the fact that they are modification sites for tubulin glutamylation and glycylation might be important for the function of microtubules. If an Glu residue that is known to be strongly glutamylated is absent in a particular tubulin isotype, this might lead to a lower glutamylation of microtubules containing this isotype.

Reviewer #2:

Remarks to the Author:

The authors have addressed my concerns. I support publication of the revised manuscript.

Reviewer #1 (Remarks to the Author):

The authors have done a great effort to answer all referees' comments. They addressed all of the concerns, and also explained why certain points could not be addressed as suggested by the referee. The paper has now strongly improved. Before being accepted for publications, some rather minor points should be addressed:

1) The introduction ends with a summary of the previously used methods to extract axonemal tubulin and explaining their problems, but there is a concluding one or two sentence missing that places the current work in this context.

We now added a short paragraph that summarize the main results of this study according to the editorial requests.

2) Figure 3: the figure is now much better, however the labelling of the lanes in panel b is missing. And as the figure does not fill the full page (and could thus be expanded), I would strongly suggest to put the antibodies and the PTMs detected next to each blot to facilitate the reading of this figure. It is very hard to go forth and back between legend and figure to find out which blot shows which PTM.

The Coomassie in panel a looks like a western blot – if the authors have a coloured picket of it, they should use this instead. It should also be stated in the legend or in the figure that this is a Coomassie-stained gel.

Finally, the diagrams in panels e-j have too small polices, could this be improved (in necessary, increase the overall size of these panels).

We changed Figure 3 according to the referee suggestions.

3) Figure 4 has improved. The polices in most panels are VERY small, and should be increased.

We increased the polices for most panels

4) The authors have done an effort to replace 'bovine tubulin' with 'bovine brain tubulin', however, there are still some places where 'bovine tubulin' is used (I detected it in the Discussion). This should be corrected.

We now changed the text from 'bovine tubulin' to 'bovine brain tubulin' in all places that we missed.

5) The authors have done a great effort to me more precise about the tubulin PTMs. I think they way they addressed the issue is satisfying. Still, there is a point that they should somehow discuss, as this is one of the most difficult technical issues in the field, and leads persistently to misleading interpretations of results. In the discussion, they write that detyr- and $\Delta 2$ -tubulin are differently present in different axonemal fractions, while tyr-tubulin is unaltered. This is technically impossible, because alpha-tubulin can be either detyr-, $\Delta 2$ - or tyr-tubulin. Thus, if detyr- and $\Delta 2$ -tubulin change, tyr-tubulin MUST also change.

Given the results presented in Fig. 3b, I would say that detyr- and $\Delta 2$ -tubulin are clearly reduced in the pellet fraction (A-tubules?), but tyr-tubulin is not. The fact that the authors cannot MEASURE an increase in tyr-tubulin relative to the total-tubulin labelling might have other reasons: some differences are very hard to see in western blot, especially, if antibodies are extremely efficient, such as the tyr-tubulin antibody. These technical issues have been extensively discussed in a method paper (Magiera & Janke 2013), and the authors must point this out to the reader to avoid further misinterpretations of their results.

We now elaborate this issue in the text. We thank the reviewer for this comment.

6) The authors might discuss the possibility that the substitution of Lys 58 to Arg has an effect on the potential acetylation of this residue. A previous whole-proteome study found Lys 58 acetylated (though this has never been confirmed by a study directly focusing on tubulin). Choudhary C, Kumar C, Gnad F, Nielsen ML, Rehman M, Walther TC, Olsen JV, Mann M (2009) Lysine acetylation targets protein complexes and co-regulates major cellular functions. Science 325: 834-840

It would be also interesting to think about the possibility that a substitution from Tyr to Phe has an effect on the potential of phosphorylating Tyr, but not Phe (in analogy with the acetylation).

This is a very interesting point, we now discuss this issue.

7) The discussion about the potential role of specific sequence motifs within the C-terminal tails of tubulin should definitely be amended by the notion that not only these primary motifs, but also the fact that they are modification sites for tubulin glutamylation and glycylation might be important for the function of microtubules. If an Glu residue

that is known to be strongly glutamylated is absent in a particular tubulin isotype, this might lead to a lower glutamylation of microtubules containing this isotype.

We added a sentence that discuss this important point.

Reviewer #2 (Remarks to the Author):

The authors have addressed my concerns. I support publication of the revised manuscript.

Jeff Moore